# A conflict between spatial selection and evidence accumulation in area LIP

Joshua A. Seideman[1], Terrence R. Stanford[1,2] & Emilio Salinas [1,2] ✉

The lateral intraparietal area (LIP) contains spatially selective neurons that help guide eye movements and, according to numerous studies, do so by accumulating sensory evidence in favor of one choice (e.g., look left) or another (look right). To examine this functional link, we trained two monkeys on an urgent motion discrimination task, a task with which the evolution of both the recorded neuronal activity and the subject's choice can be tracked millisecond by millisecond. We found that while choice accuracy increased steeply with increasing sensory evidence, at the same time, the LIP selection signal became progressively weaker, as if it hindered performance. This effect was consistent with the transient deployment of spatial attention to disparate locations away from the relevant sensory cue. The results demonstrate that spatial selection in LIP is dissociable from, and may even conflict with, evidence accumulation during informed saccadic choices.

In primates, the lateral intraparietal area (LIP) combines sensory and cognitive information to highlight behaviorally relevant locations or visual features to look at[1–3]. In simple terms, LIP 'selects' a location L when neurons with response fields (RFs) at L fire more intensely than their counterparts with RFs at other locations. Although such selection may involve many sophisticated perceptual operations[3–6], the accumulation of sensory evidence (or, more generally, temporal integration) is one of major theoretical importance. First, by some accounts[7,8], it is an obligatory antecedent to perceptually guided choices regardless of task details, sensory modality, or effector. And second, its manifestation in LIP provides key experimental justification for sequential sampling models, which comprise the most widespread computational framework for reproducing reaction time (RT) and accuracy data in deterministic choice tasks[9–12]. In this framework, the gradual differentiation between spatial locations signaled by LIP corresponds directly to the gradual formation of the perceptual decision[13,14]. So, the same neurons accumulate sensory evidence in favor of one choice or another and select a target accordingly[15].

The random-dot motion (RDM) discrimination task (Fig. 1a, b) has been pivotal to this functional interpretation. In it, the subject must look at one of two choice targets to indicate the net direction of motion of a cloud of flickering dots, and in numerous variants of the task, LIP neurons gradually signal the chosen location while

simultaneously reflecting the particulars of the perceptual discrimination[16–24]. However, in recent inactivation experiments[25,26], the LIP spatial signal distinguishing the two alternative choices was disrupted with minimal consequence to performance (effects were seen on RT but not on accuracy), consistent with a more indirect relationship between LIP activity and decision formation[14,27].

We propose an explanation for this puzzling combination of findings that is simple, consistent with LIP's role in attentional deployment[2,28,29], and yet potentially far-reaching: the perceptual evaluation of the motion stimulus occurs elsewhere and more rapidly (~200 ms) than is generally assumed, and may precede the LIP differentiation in many instances. So, what appears to be a gradual accumulation of sensory evidence is likely the byproduct of task designs that promote a slow, post-decision shift of attention from one spatial location (where the dots are) to another (where the chosen saccade target is).

This hypothesis makes a stark prediction. Consider a version of the RDM task that is urgent (Fig. 1c, d). By this, we mean that the subject must choose in a hurry, before the limited time allotted for responding expires. The details of the task will be explained later, but the main point is that the perceptual evaluation occurs while the motor planning process is already underway, so that many correct trials are rapid (low RT) but still informed by the motion stimulus. If

[1]Department of Neurobiology and Anatomy, Wake Forest School of Medicine, 1 Medical Center Blvd., Winston-Salem, NC 27157-1010, USA. [2]These authors contributed equally: Terrence R. Stanford, Emilio Salinas. ✉e-mail: esalinas@wakehealth.edu

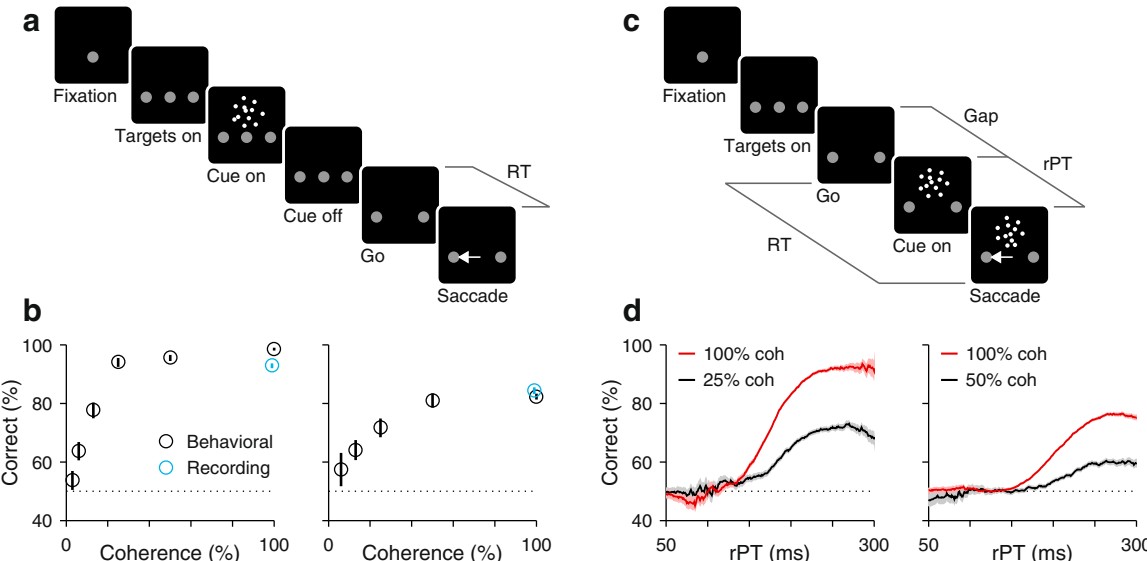

**Fig. 1 | Urgent and non-urgent motion discrimination tasks.** Subjects had to report the direction of motion (left or right) of a cloud of flickering dots by looking at one of two peripheral targets. **a** RDM task (non-urgent). The motion stimulus is presented and evaluated first (between Cue on and Cue off, 600–1000 ms), well before the go signal (fixation point offset; Go). The time interval between the go and saccade onset (RT) is indicated. **b** Performance in the RDM task as a function of motion coherence. Results are from behavioral (black circles) and recording sessions (cyan circles) from monkeys C (left; 7363 behavior trials; 8685 recording trials) and T (right; 4547 behavior trials; 3952 recording trials). Points indicate percentage of correct choices across trials and error bars indicate 95% confidence intervals. **c** CRDM task (urgent). The go signal (Go) is presented first and the motion stimulus follows (Cue on) after an unpredictable delay period between them (Gap, 0–250). The RT time window for responding is limited (350–425 ms), so the perceptual evaluation must occur as the motor plan develops. The likelihood of success is dictated by the processing time interval (rPT = RT − gap), which is when the motion cue is visible. Gray lines mark intervals (RT, Gap, rPT) between events (black squares). **d** Performance in the CRDM task as a function of rPT, or tachometric curve. Results are from behavioral sessions from monkeys C (left) and T (right) for 100% (red; C: 9544, T: 33,971 trials) and a lower coherence (black; C: 7,909, T: 12,066 trials). Each point includes trials within a 51 ms rPT bin. Traces show percentage of correct choices across trials in each bin, and shaded error bands indicate ± 1 SE. Source data are provided as a Source Data file.

LIP neurons accumulate evidence, then in those trials they must still differentiate and indicate the impending choice, with stronger evidence yielding stronger differentiation. Alternatively, if the spatial differentiation in LIP occurs after the motion stimulus has been evaluated, its development on such rapid trials will be curtailed, and stronger evidence will not prevent its attenuation or abolition altogether.

## Results

### Urgent versus non-urgent choices
To test this prediction, we recorded single-neuron activity in area LIP during two variants of the RDM discrimination task (Methods). In the standard, non-urgent version (Fig. 1a), the motion stimulus is presented first (for 600–1000 ms) and is followed by the offset of the fixation point (Go), which means "respond now!" In the urgent or compelled random-dot motion (CRDM) discrimination task (Fig. 1c), the order of events is reversed: the go signal is given first, before the stimulus is shown, and the subject must respond within a short time window after the go (350–425 ms). Although the required perceptual judgment is the same, the tasks differ critically in the order in which perceptual and motor processes are engaged. In the former, the saccade can be prepared with relative leisure, after the perceptual evaluation is completed, whereas in the latter, the motor plan is initiated early and the perceptual evaluation must occur while the developing motor plan advances. Under time pressure, saccades can be triggered before, during, or shortly after the perceptual evaluation, and may result in guesses, partially informed, or fully informed choices (Fig. 1d). Exactly which of these outcomes is observed depends on a quantity that we call the raw processing time (rPT), which is the amount of time during which the cue can be seen and analyzed (computed as RT − gap in each trial; Fig. 1c). As elaborated below, this is the fundamental variable in the task. This way, perceptual and motor performance (RT) still exhibit conventional dependencies on task difficulty

(Supplementary Fig. 1), but are effectively decoupled[30–32] (Supplementary Fig. 2).

Two monkey subjects performed the two choice tasks in interleaved blocks of trials (in addition to single-target tasks traditionally used to characterize LIP activity; Fig. 2a, b). In the standard, non-urgent RDM task, most choices were correct (93% and 84% correct for monkeys C and T at 100% coherence; Fig. 1b), and the recorded LIP activity evolved as reported previously[16,17,20,23] (Fig. 2c). The neurons responded briskly upon presentation of a choice target in the RF, continued firing at an elevated rate, and began signaling the choice about 200 ms after the onset of the motion stimulus (Fig. 2d, red arrow), at which point their activity increased for saccades into the RF and decreased for saccades away.

To interpret this growing differential signal (quantified by $S_{ROC}$, Fig. 2d) as an immediate correlate of the perceptual evaluation—one that is causal to the choice—one must assume that the evaluation begins about 200–250 ms after cue onset. And indeed, many experiments are consistent with such a protracted time scale[9,18–20,22,23]. However, none of these studies tracked the time course of performance explicitly, moment by moment (Supplementary Note 1). By doing this, we find that after 250 ms of stimulus viewing time the motion discrimination is essentially over, as detailed next.

### Neural discrimination conflicts with motion discrimination under time pressure
As mentioned, the key variable in the CRDM task is the rPT, the amount of time during which the stimulus is available for processing before movement onset (Fig. 1c). Plotting choice accuracy as a function of rPT yields a detailed, high-resolution account of the temporal evolution of the perceptual judgment (Figs. 1d and 2h). According to this 'tachometric' curve, in trials with rPT ≲ 140 ms the stimulus is seen so briefly that the motion direction cannot be resolved, which results in uninformed choices, or guesses (~50% correct). Choice accuracy then rises

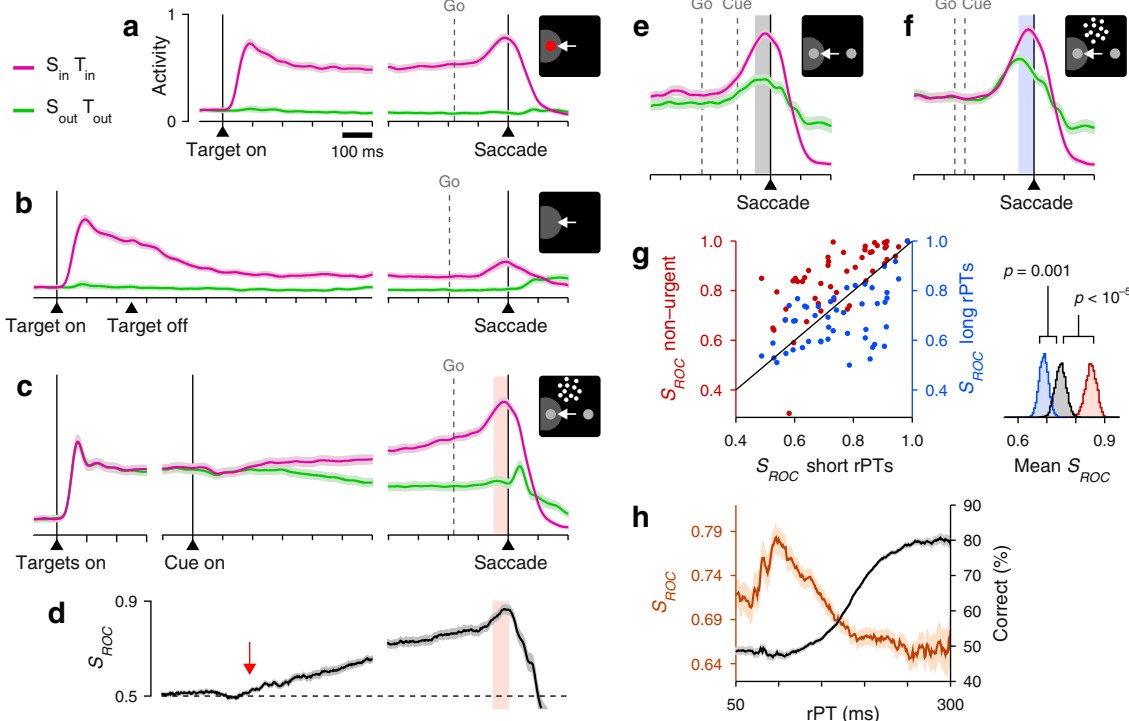

**Fig. 2 | LIP activity in urgent versus non-urgent random-dot motion discrimination. a** Responses during visually guided saccades. Traces show normalized firing rate (mean ± 1 SE across cells; $n = 50$) as a function of time for correct trials into (magenta) or away from the cell's RF (green). Same axes for panels **b**, **c**, **e**, **f**. The gray dotted line indicates the median onset of the go signal. **b** Responses during memory-guided saccades ($n = 49$). **c** Responses in the non-urgent RDM task ($n = 51$). **d** Spatial signal magnitude (mean ± 1 SE) as a function of time for the data in **c** (same time axis). Throughout the article, $S_{ROC}$ measures the statistical separation between inward and outward responses (Methods). Red arrow marks approximate onset of differentiation (190 ms). **e** Responses in the CRDM task ($n = 51$) during guesses (rPT ≤ 150 ms). **f** Responses in the CRDM task during fully informed choices (rPT ≥ 200 ms). **g** Presaccadic $S_{ROC}$ for individual neurons ($n = 51$) in the non-urgent RDM task (left $y$-axis), and during guesses ($x$-axis) and fully informed choices in the CRDM task (right $y$-axis). Spike counts for computing $S_{ROC}$ are from shaded windows in **c–f**. For each condition, the side plot shows the bootstrapped distribution of the mean $S_{ROC}$ across neurons, with $p$ values for differences evaluated via one-sided permutation tests (Methods). **h** Behavioral (black) and neuronal (brown) performance curves (mean ± 1 SE across trials) from the same CRDM sessions. $S_{ROC}$ is from presaccadic spikes pooled across neurons ($n = 51$) and sorted by rPT (bin width = 51 ms). All data are from correct trials, except for the short rPTs in **g**, which combine correct and incorrect trials. All motion data are for 100% coherence. Source data are provided as a Source Data file.

rapidly after the 150 ms mark, reaching asymptotic performance for rPTs of 200–250 ms. This amount of processing time is sufficient for evaluating the RDM stimulus and reliably determining its motion direction.

As in other urgent tasks with similar designs[30–33], the rPT measured in each trial quantifies the degree to which sensory evidence guided the corresponding choice (or the probability that the choice was guided). Thus, if the differential signal in LIP reflects the amount of evidence accumulated in each trial, then it should be larger for fully informed discriminations (at long rPTs) than for guesses (at short rPTs), and its evolution should parallel the rise of the tachometric curve.

Contrary to this expectation, the recorded LIP activity showed quite the opposite. During performance of the CRDM task, the neural responses favoring each of the two possible eye movements were clearly separated just prior to saccade onset (Fig. 2e, f). This separation was quantified by contrasting the numbers of spikes elicited by saccades into the RF versus saccades away in the 50 ms preceding movement onset (Fig. 2c–f, shaded areas; Methods). The resulting presaccadic separation ($S_{ROC}$) was less definitive in the urgent condition than in the non-urgent (Fig. 2g, red data), but the urgent differential signal still pointed reliably to the eventual choice. Crucially, however, across the sample of individual neurons recorded in the CRDM task ($n = 51$), the differential signal measured during fully informed, correct choices (rPT ≥ 200 ms) was considerably weaker than that during guesses (rPT ≤ 150 ms; Fig. 2g, blue data, $p = 0.001$, permutation test). More evidence yielded less

differentiation. Furthermore, when the presaccadic responses were pooled across neurons and binned by rPT to assess how the spatial signal develops as a continuous function of processing time (Methods), the resulting neurometric curve decreased steadily for rPT > 100 ms (Fig. 2 h, brown curve)—in sharp contrast to choice accuracy (Fig. 2h, black curve). In the CRDM task, the stronger the influence of perception on the choice, the weaker the observed LIP differentiation.

**Stronger LIP differentiation predicts higher error probability**
Everything else being equal, the neural encoding of perceptual information upon which choices are made is typically more robust for correct than for incorrect outcomes[34–37]. This is true across tasks, circuits, and modalities, and should apply to urgent choices too. We therefore examined the LIP selection signal, i.e., the difference in presaccadic activity between movements into the RF (Fig. 3a, positive bars) and movements away (Fig. 3a, negative bars), in correct and error trials. To maximize statistical resolution, for this analysis we first pooled the data across neurons (Methods).

During short-rPT trials (rPT ≤ 150 ms), the responses in LIP were identical for correct (Fig. 3a, first two gray bars) and incorrect choices (Fig. 3a, last two gray bars), as anticipated given that those were all guesses. Consequently, the differential signals contrasting activity into versus away from the RF (i.e., $S_{ROC}$ separation between positive and negative bars in Fig. 3a) were the same for correct and incorrect eye movements in this case. During informed discriminations (rPT > 150 ms), however, the differential signal was greater for errors (Fig. 3a, b,

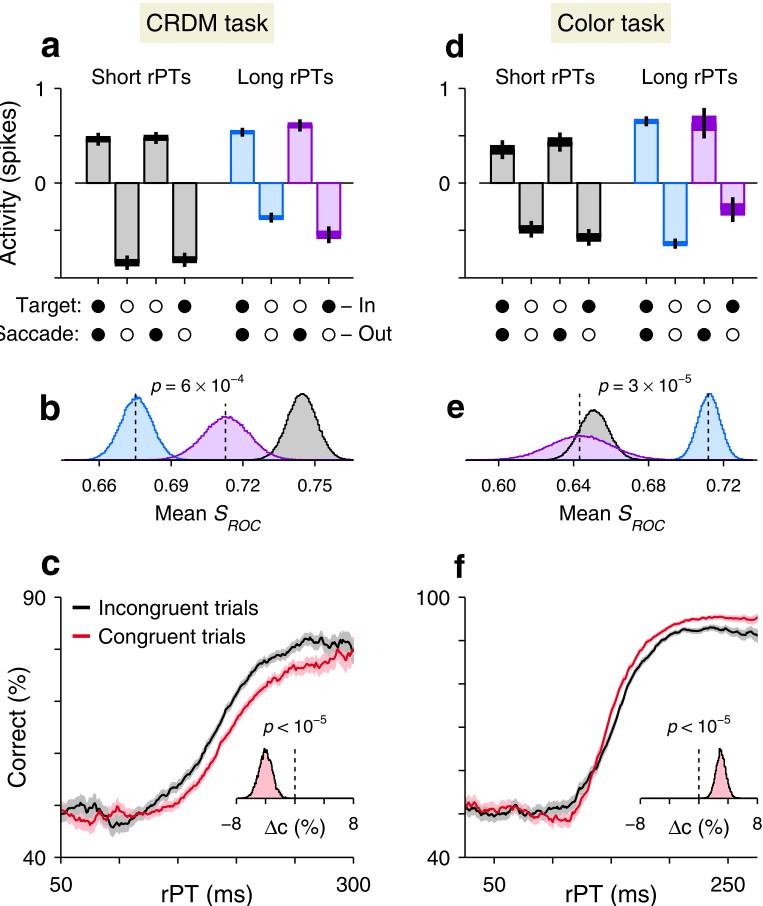

**Fig. 3 | LIP differentiation may seem to help or hinder performance. a** LIP activity in the CRDM task during guesses (rPT ≤ 150 ms, gray) and informed choices (rPT > 150 ms, blue, purple) sorted by outcome (x-axis). Circles indicate whether target and saccade were in (filled) or out of the RF (open) in each case. Activity indicates spike counts (from 50 ms presaccadic window) centered for each neuron (mean subtracted; Methods) and pooled across neurons (n = 51). Bars indicate mean values, widths of horizontal lines indicate 68% confidence intervals, and vertical lines indicate 95% confidence intervals. Underlying numbers of trials are 1187–3704 (range) across bars. **b** Differential signal magnitudes for the three conditions in **a** indicated by color. For guesses, both correct and incorrect trials are included. Curves are bootstrapped distributions. The indicated p-value is from a one-sided resampling test (Methods). **c** Performance in the CRDM task conditioned on neuronal activity. Trials were classified according to their presaccadic spike counts as either congruent (red) or incongruent (black) with strong differentiation. Traces show percentage of correct choices across trials in each rPT bin, and shaded error bands indicate ± 1 SE. Inset shows bootstrapped distribution for the mean difference in percent correct between curves for rPTs of 130–230 ms. The indicated p-value is from a one-sided resampling test (Methods). **d–f** As in **a–c**, but for the urgent color-discrimination task (n = 56). In **d**, underlying numbers of trials are 467–3855 (range) across bars. In **d**, **e**, rPT ≤ 125 ms for guesses and rPT > 125 ms for informed choices. In **f**, difference was evaluated for rPTs between 140–280 ms. Source data are provided as a Source Data file.

purple data) than for correct choices (Fig. 3a, b, blue data; p = 0.0006, resampling test)—again, opposite to the trend expected from an evidence accumulation process.

In urgent tasks, the relationship between behavioral performance and single-neuron activity is revealed most effectively by conditioning the former on the latter. First, for a given experimental condition (saccade into or away from the RF), the spike counts collected from a neuron are sorted by magnitude (above vs. below the median), and then performance is compared across the corresponding groups of trials (Supplementary Fig. 3; Methods). The resulting tachometric curves conditioned on evoked activity reveal if, when, and how the subject's behavior changes when the recorded neurons fire more or less than average. According to this analysis, performance was comparatively poor (p < 10⁻⁵, resampling test) in trials that were congruent with strong spatial differentiation, when saccades into the RF yielded high spike counts or when saccades away yielded low counts (Fig. 3c, red trace). Conversely, performance was comparatively better in trials that were incongruent with strong spatial differentiation, when saccades into the RF yielded low spike counts or when saccades away yielded high counts (Fig. 3c, black trace; see Supplementary Fig. 3a–c for individual RF conditions). The relative shift between the congruent

and incongruent curves means that, when the LIP spatial signal was strong, more processing time was needed to achieve a given accuracy than when the signal was weak. This is as if a more robust spatial signal interfered with the urgent motion discrimination.

## Spatial conflict within LIP

Why is the LIP differentiation suppressed in the CRDM task, and more so for informed choices? Two possible reasons stand out, both brought about by urgency and both likely, given LIP's participation in attentional deployment[2,28,29]. First, the differential signal is curtailed when it has less time to develop (Fig. 2g, red data), a general effect[33] consistent with our initial hypothesis (that, time permitting, LIP selection is subsequent to the perceptual evaluation). And second, the particular geometry of the task must create a spatial conflict: the early motor plan initiated shortly after the go signal[30,33] automatically allocates attentional resources to the planned saccade endpoint(s)[38–42], but attention should be directed to the RDM stimulus, which defines the perceptually relevant location[25,26]. A spatial competition ensues[28,43].

Evidence of this is plainly manifest in the behavioral CRDM data, which show that saccades are briefly but almost completely suppressed shortly after the onset of the dots (Fig. 4a). This suppression is

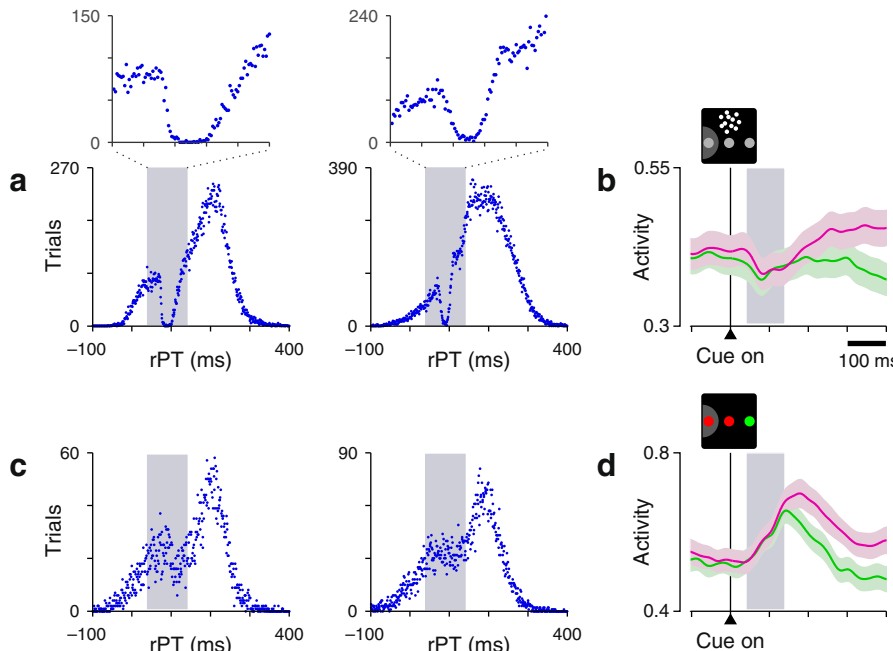

**Fig. 4 | Evidence of spatial conflict in the CRDM task. a** Processing time distributions from the CRDM behavioral sessions of monkeys C (left, 30,473 trials) and T (right, 52,945 trials). Histograms were computed using non-overlapping 1 ms bins, so each point indicates the number of trials obtained with a given integer rPT. The data comprise correct and error trials from all coherences. Insets zoom in on the rPT range of 40–140 ms. Note the nearly complete absence of events at 80–100 ms. **b** Neuronal response to the motion onset in the non-urgent RDM task. Traces show normalized firing rate (mean ± 1 SE across cells) as a function of time for correct trials into (magenta) or away from the cell's RF (green). Same data as in Fig. 2c, but restricted to a period around cue onset. **c** As in **a**, but for the urgent color-discrimination task. The data comprise all correct and error trials from monkeys C (left, 7330 trials) and T (right, 10,734 trials). **d** As in **b**, but for the non-urgent color-discrimination task. Same data as in Fig. 6c, but restricted to a period around cue onset. All gray shaded areas mark the 40–140 ms interval following cue onset. Source data are provided as a Source Data file.

consistent with two well-documented oculomotor phenomena triggered by salient stimulus onsets, the exogenous capture of attention[28,32,44,45] and the inhibition of impending saccades[45–47], and its timing (~90 ms after cue onset) coincides with a slight decrease in LIP activity (Fig. 4b) often observed in the non-urgent RDM task[16,17,20,23]. The motion-driven response is in intense conflict with the oculomotor activity that generates saccades to the choice targets, at least initially.

## LIP's apparent contribution depends on stimulus-choice configuration

To investigate how perceptual performance and LIP selection depend on these two factors, limited time and attentional conflict, we recorded LIP activity from the same monkeys during two versions, urgent and non-urgent, of a discrimination task in which the subject must make an eye movement to the peripheral stimulus that matches the color of the fixation point[30,31,33] (Fig. 5). The key difference here is that the conflict described above is eliminated: the relevant color cues are found at the choice targets, and deploying attention/perceptual resources to them should be of benefit, if not a necessity, to the required discrimination.

Indeed, consistent with this logic, saccades were minimally suppressed in this configuration (Fig. 4c), as expected from the abrupt cue onset occurring at the two goal locations rather than at a third, non-goal location[30,43,45,47]. Furthermore, the transient, undifferentiated response to the cue onset was an increase in the activity aligned with the choice targets (Fig. 4d), rather than a decrease (Fig. 4b).

Importantly, during the non-urgent color-matching task, the sampled neurons (which again exhibited characteristic LIP response features; Fig. 6a, b) also differentiated saccades into versus away from the RF (Fig. 6c, d). The differential signal rose above chance slightly earlier in the color task than in the standard RDM task (Figs. 2d and 6d, arrows), but it achieved the same magnitude just before saccade onset (in both tasks the presaccadic $S_{ROC}$ was $0.85 ± 0.02$, mean ± SE across

cells). Overall, under relaxed, non-urgent conditions, the evoked spatial signal developed with comparable timecourse and strength in the motion- and color-based tasks, in spite of their distinct spatial and feature requirements. Under time pressure, though, the comparison across tasks was striking. During the urgent color-matching task, the differential response in LIP was larger for informed than uninformed discriminations (Fig. 6e–g); its magnitude increased over time in parallel with the monkeys' choice accuracy (Fig. 6 h); it was weaker for errors than correct choices during informed trials (Fig. 3d, e); and it acted as if to improve the monkeys' performance (Fig. 3f). In this case, the greater the influence of perception on the choice, the stronger the spatial signal observed in LIP.

These results in the color-matching experiment confirm that an informed spatial signal can emerge very rapidly in LIP[48,49]. They show that time pressure alone does not necessarily abolish or reverse the expected correlation between sensory evidence and LIP differentiation. Therefore, urgency alone cannot explain the CRDM results. Rather, the data suggest that the anticorrelation between CRDM performance and LIP spatial signal strength results from urgency exacerbating a spatial conflict between the perceptually relevant location and the saccade endpoint (see "Discussion").

## Potential pitfalls

Notably, an early bias favoring choices into the RF is visible in the CRDM data (Fig. 2e), but this simply reflects a consistent preference in the initial guess that is required of the subjects in every urgent trial. Such consistency is of little consequence to the perceptual evaluation[21,30]. Indeed, the results did not change qualitatively when this bias was eliminated on a trial-by-trial basis (Supplementary Fig. 4), nor when it was either enhanced or suppressed by suitable selection of experimental sessions (Supplementary Fig. 5) or recorded trials (Supplementary Fig. 6). Also, for both the motion- and color-based tasks the

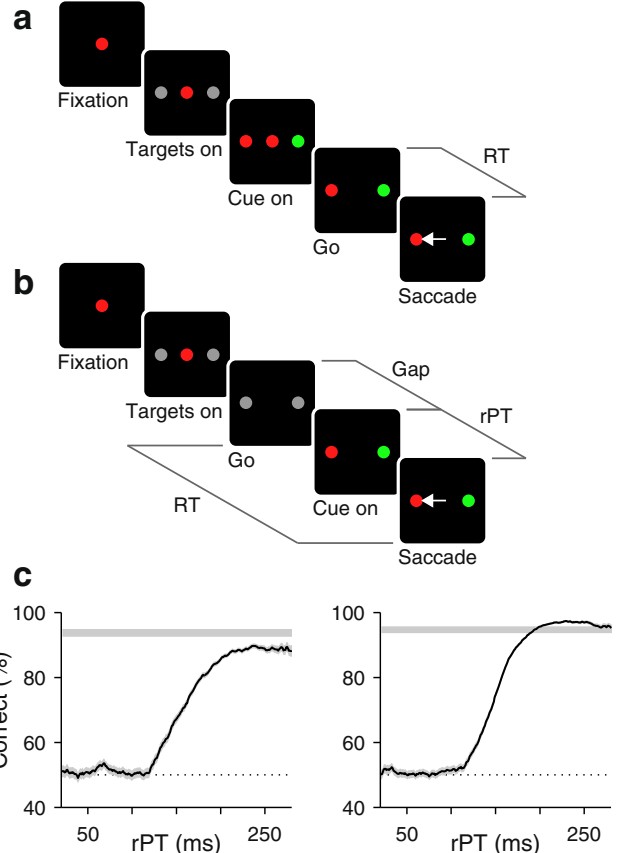

**Fig. 5 | Urgent and non-urgent color-discrimination tasks.** Subjects had to look to the peripheral target that matched the color of the fixation point. **a** Easy, non-urgent task. The color stimuli are presented first (Cue on), and after 500–1000 ms, the go signal (fixation point offset; Go) is given. **b** Urgent task. The go signal is given first, before the cue is revealed. All time parameters and intervals are as in the CRDM task (Fig. 1c). **c** Tachometric curves based on all the recording sessions during which the urgent color-discrimination task was performed by monkeys C (left, 7330 trials) and T (right, 10,745 trials). Each point includes trials within a 51 ms rPT bin. Traces show percentage of correct choices across trials in each bin, and shaded error bands indicate ±1 SE. Gray horizontal strips indicate performance in the non-urgent task (95% confidence interval). Source data are provided as a Source Data file.

results were robust with respect to the subjects' performance level (Supplementary Fig. 7), the criteria used for including/excluding neurons (Supplementary Figs. 8 and 10), and how the effects were quantified (Supplementary Fig. 11). Finally, the results were minimally affected by history effects (Supplementary Fig. 12).

## Discussion

The highly robust target selection seen during non-urgent conditions (RDM task) would lead one to conclude, as have countless past studies, that LIP differentiation is an obligatory, causal antecedent to perceptually informed choices, and that greater differentiation implies more or stronger perceptual evidence. Yet, for equally informed choices made urgently (CRDM task), the spatial signal was markedly attenuated, it decreased with increasing evidence, and appeared to hinder performance. These findings demonstrate that LIP serves a distinct visuomotor function dissociable from—and at times incompatible with—the accumulation of sensory evidence.

Both the effects of inactivation[25,26] and our CRDM results may seem counterintuitive; but why are the latter so extreme, with LIP activity pointing more strongly to erroneous than correct choices, and more weakly to informed than uninformed ones? Our unique

paradigm creates a scenario in which such outcomes seem extraordinary, but the interpretation is entirely consistent with what is known about attention coding in LIP.

Our results stem from a visuomotor conflict that requires three conditions to become apparent: (1) tight coupling between spatial attention, understood as a mechanism for enhancing perceptual judgments at specific locations, and saccade planning, (2) a spatial geometry in which the relevant sensory stimulus is away from the potential saccade targets, and (3) time pressure. The first condition is well established[38–42]; the necessity of the second is clear from our results in the urgent color-discrimination task, and is supported by experiments in which endogenous (voluntary) and exogenous (stimulus-driven) attention are dissociated[29,32,42,44]; and the necessity of the third is obvious from the comparison between the urgent and non-urgent motion tasks. With these conditions in place, a plausible outline of the dynamics of attention in the CRDM task would be as follows.

Because the go signal is given first, an eye movement is planned early on[30,31], and this automatically commits attentional resources to one or both choice targets and away from the location of the dots[38–42]. Thus, the stronger the commitment of the uninformed motor plan, the less attentional resources can be deployed to the dots, and the lower the likelihood that the choice will be correct. Previous neurophysiological studies fit with this account: the LIP circuitry is inherently competitive[43], its differential activity encodes where attention is directed to[2,28], and performance in the RDM task is substantially impaired when the LIP neurons with RFs covering the dots are inactivated[26], indicating that those neurons are relevant to the perceptual evaluation of the motion stimulus. Within this competitive scheme, interpretation of our neural data is fairly straightforward: the early selection of a saccade target would correspond to attention being diverted away from the location of the dots, consistent with a negative correlation between LIP differentiation and performance during a brief but critical period of time when the motion stimulus is being evaluated (rPT ≈ 100–250 ms). This is best illustrated by the behavioral curves conditioned on neuronal activity (Fig. 3c, Supplementary Fig. 3a–c), because they show that relatively strong differentiation leads to relatively poor performance for any fixed amount of processing time (in the informed range)—presumably because attention on the dots is always relatively reduced.

We stress that time pressure is critical here. When the urgency requirement is relaxed (standard RDM task), attention can be deployed to the location of the dots even before motion onset, and can remain there as long as necessary. The focus on the dots need not be long, ~200 ms, considering the time to approach asymptotic performance (Figs. 1d and 2h), and once the perceptual evaluation nears completion, attention can shift to the appropriate choice target as the response saccade is planned. Thus, from the perspective of an LIP neuron covering one of the choice targets, this transition will look like a single, gradual, monotonic process that starts ~200 ms after motion onset (Fig. 2d, arrow), and because its timecourse and magnitude may still depend on the strength of the sensory evidence, the resulting post-perceptual differentiation may appear causal to the choice.

This interpretation is in line with the early observation[16] that the rate at which the LIP differential signal diverges during the RDM task depends on the monkey's expectation of the stimulus duration, and with more recent analyses[27] showing that, although LIP may encode both sensory evidence and time-varying premotor buildup, these signals are dissociable and independent of each other. It also provides a plain explanation for the outcome of the inactivation experiments[25,26]: when the neurons with RFs at one of the choice targets are silenced, no effect on accuracy is observed because those cells do not actually accumulate evidence, they simply appear to do so when attention shifts to the chosen target; disrupting this shift simply delays the saccade, which increases the RTs; and when the silenced neurons are

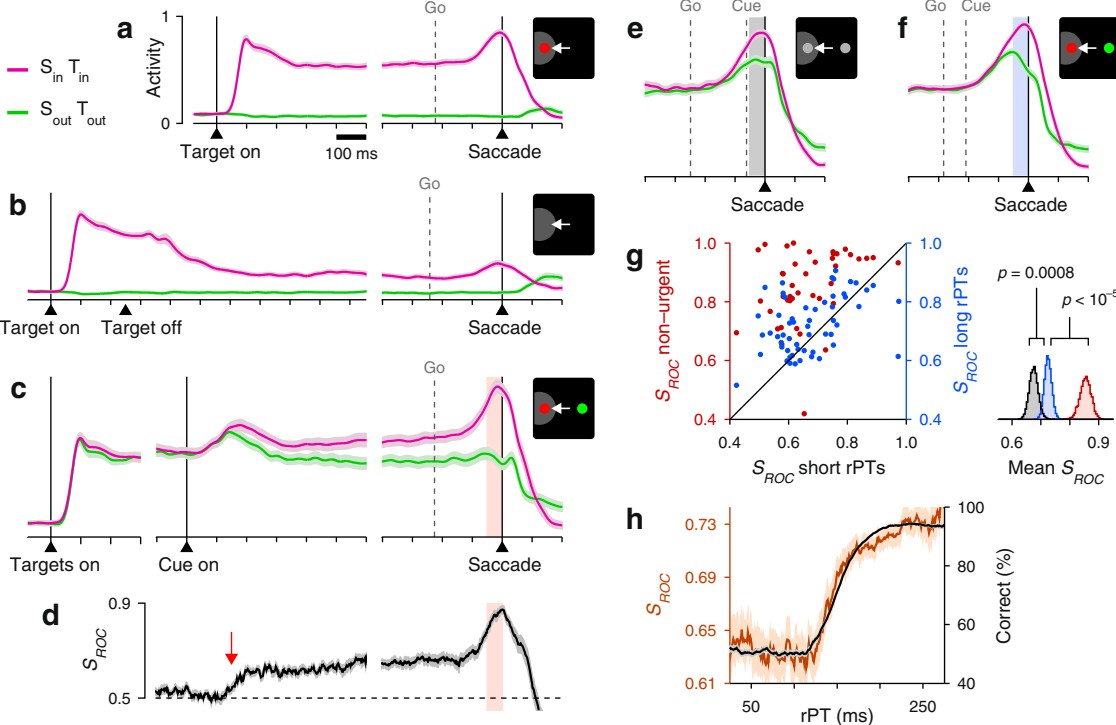

**Fig. 6 | LIP activity in urgent versus non-urgent color discrimination.** For all panels, the shown quantites and format are the same as in Fig. 2. In **a**, **b**, **e**–**h**, data are from $n = 56$ sampled neurons. In **c**, **d**, **g** (red points), data are from

$n = 43$ sampled neurons. In **d**, arrow is at 150 ms. In **e**, rPT ≤ 125 ms. In **f**, rPT ≥ 175 ms. Source data are provided as a Source Data file.

those with RFs overlapping the dots[26], the motion discrimination becomes harder because attention is disrupted at the critical location.

This account is not complete, though. It is unclear how exactly LIP activity maps onto attention that is divided three ways. And although much is known about attention in tasks that impose prolonged fixations and long delays, the neural correlates of attentional selection are likely to be more subtle and evolve more rapidly for saccadic choices that are decided within the 200–250 ms time frame that characterizes natural viewing conditions[29,32,33,44,45,47]. In addition, when a saccade is imminent, the RFs of LIP neurons shift toward the saccade endpoint[50], and it is unclear whether such shifts correspond directly to displacements of attention or to distinct, complementary phenomena–although in either case, they would likely result in diminished perceptual resources at the location of the dots. Regardless of these gaps, however, the current data are broadly consistent with area LIP's well cemented role in attentional deployment, and show that the spatial signal it generates is not an obligatory correlate of perceptual evidence during informed saccadic choices.

## Methods
### Subjects and setup
All experimental procedures were conducted in accordance with NIH guidelines and USDA regulations, and were approved by the Institutional Animal Care and Use Committee (IACUC) of Wake Forest School of Medicine. The subjects in this experiment were two adult male rhesus monkeys (*Macaca mulatta*) weighing between 8.5 and 11 kg. For each animal, an MRI-compatible post (Crist Instruments, MD, USA) was implanted on the skull while under general anesthesia. The post served to fix the position of the head during all experimental sessions. Following head-post implantation, both subjects were trained to perform oculomotor response tasks in exchange for water reward. After reaching a criterion level (>75% accuracy for each task), craniotomies were made and recording cylinders (Crist Instruments, MD, USA) were placed over the LIP of each monkey (monkey C: left hemisphere;

monkey T: left and right hemispheres; stereotactic coordinates: 5 mm posterior, 12 mm lateral[51,52]) while under general anesthesia. Neural recordings commenced after a 1–2 week recovery period following cylinder placement.

### Behavioral and neurophysiological recording systems
Eye position was monitored using an EyeLink 1000 Plus infrared tracking system (SR Research; Ottawa, Canada) at a sampling rate of 500 or 1000 Hz. For sessions in which dot-motion tasks were performed, all gaze-contingent stimulus presentation and reward delivery were controlled using Psychtoolbox[53,54] version 2.0 (publicly available); for all other sessions, gaze-contingent stimulus presentation and reward delivery were controlled via a custom-designed PC-based software package from Ryklin Software (2016 version). Visual stimuli were presented on a Viewpixx/3D display (Vpixx Technologies, Quebec, Canada; 1920 × 1080 screen resolution, 120 Hz refresh rate, 12 bit color) placed 57 cm away from the subject. Viewing was binocular. During task performance, the radius of the fixation and target windows was approximately 3 degrees of visual angle. Red and green spots were isoluminant (23.5 cd/m²). For the dots tasks, the fixation and target spots were 1.0 and 1.5 degrees of visual angle, respectively. For all other tasks, the fixation and target spots were 1.7 degrees.

For the motion stimuli, the dots were 3 × 3 pixels each and were presented within a circular aperture of 5 degrees on the center of the screen or just above the fixation point. The motion was generated with the same algorithm described previously[17,55], which is based on three independent frames with dots. The three frames cycle sequentially. Every time a frame is replotted, a fixed percentage of its dots are displaced in the same direction and the rest of the dots are relocated randomly. The dots that are displaced coherently are selected randomly every time a frame is updated and produce motion; the remaining, non-coherent dots produce no net motion. In practice, our implementation of the algorithm was slightly modified so that the resulting motion in our setup (running at 120 Hz refresh rate) appeared

the same as with the standard algorithm in a conventional setup (at 60 Hz).

Neural activity was recorded using single tungsten microelectrodes (FHC, Bowdoin, ME; 2–4 MΩ impedance at 1 kHz) driven by a hydraulic microdrive (FHC). A Cereplex M headstage (Blackrock Microsystems, UT, USA) filtered (0.03 Hz to 7.5 kHz), amplified, and digitized electrical signals, which were then sent to a Cereplex Direct (Blackrock Microsystems) data acquisition system. Single neurons were isolated online based on amplitude criteria and/or waveform characteristics.

## Behavioral tasks

Three design elements are the same for all the tasks. (1) Each trial begins with presentation of a central spot and the monkey fixating it for 300–800 ms. (2) The offset of the fixation spot is the go signal that instructs the monkey to make a saccade. (3) To yield a reward (drop of liquid), the saccade must be to the correct location and must be initiated within an allotted RT window. The RT is always measured as the time elapsed between fixation offset and saccade onset (equal to the time point following the go signal at which the eye velocity first exceeds a criterion of 25 degrees/s). In non-urgent tasks the monkey is allowed to initiate an eye movement within 600 ms of the go signal, whereas in urgent tasks this must happen within 350–425 ms.

**Visually- and memory-guided saccade tasks.** Two standard single-target tasks were used to characterize the visuomotor properties of LIP neurons. In both tasks, after the monkey fixates, a peripheral target is presented (Target on) either within or diametrically opposed to the RF of the recorded neuron. For the delayed visually guided saccade task, after a variable delay (500–1000 ms), the fixation spot disappears (Go) and the monkey is required to make a saccade to the peripheral target. For the memory-guided saccade task, after being displayed for 250 ms, the peripheral target is extinguished (Target off) and the monkey is required to maintain fixation throughout a subsequent delay interval (500–1000 ms). After this memory interval, the fixation spot disappears (Go) and the monkey is required to make a saccade to the remembered target location.

**Non-urgent RDM motion discrimination task.** This two-alternative task (Fig. 1 a) is similar to previous implementations of the RDM discrimination task[16,17,20,23]. Upon fixation and after a short delay (300–500 ms), two gray stimuli, the potential targets, are presented (Targets on), one in the RF and one diametrically opposed. After a delay (250–750 ms), a cloud of randomly moving dots appears in the center of the screen (Cue on) or just above the fixation point; the motion lasts 600–1000 ms (until Cue off). Then, after another delay period (300–500 ms), the fixation spot is extinguished (Go), which instructs the monkey to make a choice. If the saccade is to the stimulus in the direction of the dot motion and is made within 600 ms, the monkey obtains a liquid reward. The direction of motion, toward one choice target or the other, is assigned randomly from trial to trial. The difficulty of the task varies with stimulus coherence, which is the percentage of dots that move in a consistent direction across video frames. Monkeys worked with coherence values of 100%, 50%, 25%, 6% and 3%, but the neural data were recorded at 100% (Fig. 1a, b).

**Compelled random-dot motion discrimination task.** The CRDM task (Fig. 1c) is an urgent version of the RDM discrimination task just described. The geometry, reward size, and stimuli are the same; only the temporal requirements are different. In this case, the monkey fixates, the two peripheral gray stimuli are shown (Targets on), and after a delay (250–750 ms), the go signal is given (Go), urging the subject to respond as quickly as possible (within 350–425 ms). At this point in the trial, however, no information is available yet to guide the choice. That

information, conveyed by the cloud of flickering dots, is revealed later (Cue on), after an unpredictable amount of time following the go (Gap; 0–250 ms). Subjects are tasked with looking to the peripheral choice alternative that is congruent with the net direction of motion of the dots (Saccade).

On each trial, the raw processing time, or rPT, is the maximum amount of time that is potentially available for seeing and evaluating the motion stimulus. It is the time interval between cue onset and saccade onset (rPT = RT – gap). We refer to it as 'raw' because it includes any afferent or efferent delays in the circuitry[30]. Gap values (0–250 ms) varied randomly from trial to trial and were chosen to yield rPTs covering the full range between guesses and informed choices.

**Non-urgent color-discrimination task.** In this task (Fig. 5a), the color of the central fixation spot (red or green) defines the identity of the eventual target. Upon fixation and after a short delay (300–800 ms), two gray stimuli, the potential targets, are presented (Targets on), one in the RF and one diametrically opposed. After a delay (250–750 ms), one of the gray stimuli changes to red and the other to green (Cue on). After a cue viewing period (500–1000 ms), the fixation spot is extinguished (Go), which instructs the monkey to make a choice. If the ensuing saccade is to the stimulus that matches the color of the prior fixation spot and is made within 600 ms, the monkey obtains a reward. Colors and locations for target and distracter are randomly assigned in each trial.

**Urgent color-discrimination task.** This task (Fig. 5b), also referred to as the compelled-saccade task[30,31,33], requires the same red-green discrimination as in the easier non-urgent version. In this case, after the monkey fixates (300–800 ms) and the two gray stimuli in the periphery are displayed (Targets on; 250–750 ms), the fixation spot disappears (Go). This instructs the monkey to make a choice, although the visual cue that informs the choice (one gray spot turning red and the other green; Cue on) is revealed later, after an unpredictable period of time following the go signal (Gap; 0–250 ms). To obtain a reward, the monkey must look to the peripheral stimulus that matches the color of the initial fixation spot (Saccade) within the allowed RT window (350–425 ms). As with the CRDM task, the key variable that determines performance is the rPT.

## Tachometric curves and rPT intervals

All data analyses were performed in Matlab (The MathWorks, Natick MA). To compute the tachometric curve and rPT distributions, trials were grouped into rPT bins (width equal to 1 ms in Fig. 4, 51 ms elsewhere), with bins shifting every millisecond. Numbers of correct and incorrect trials were then counted within each bin. From these numbers, we calculated the percentage of correct choices and, using binomial statistics, error bars and confidence intervals for the percentage.

To parse trials into short and long-rPT time bins (Figs. 2e–g, 3a, b, d, e and 6 e–g), we considered the distributions of processing times from all the recording sessions in each task. The threshold for guesses (rPT ≤ 150 for the CRDM task; rPT ≤ 125 ms for the color task) corresponded to the point at which the fractions of correct and incorrect trials started diverging steadily with rPT. We distinguish between informed choices, which were all the trials above this cutoff, and fully informed choices, which were the trials above this cutoff plus 50 ms, which brought the fraction correct about 75% of the way from chance to asymptotic. The results depended minimally on the exact cutoffs used.

Tachometric curves conditioned on neuronal activity (Fig. 3c, f) were computed as follows. First, for each neuron, spike counts from a presaccadic window (−50:0 ms, aligned on saccade) were collected and sorted into two conditions, saccade-in ($S_{in}$) and saccade-out ($S_{out}$) choices. The trials in each condition were then split into

two groups, with spike counts below the median for the condition, or with spike counts at or above it. Four groups of trials resulted: $S_{in}$ high firing, $S_{in}$ low firing, $S_{out}$ high firing, and $S_{out}$ low firing. Data from all the neurons in a sample were aggregated, and a tacho-metric curve was generated for each group (Supplementary Fig. 3). The first and last groups are congruent with a strong spatial signal, whereas the other two are incongruent. Because the results were consistent for $S_{in}$ and $S_{out}$ conditions (Supplementary Fig. 3), trials were combined across these to produce a single congruent data set and a single incongruent one.

For the CRDM data, differences between tachometric curves conditioned on low versus high firing were quantified and evaluated for significance (see below) for rPTs of 130–230 ms. This same range was used for all such analyses regardless of how the data were parsed. For the urgent color-discrimination data, the corresponding range was 140–280 ms.

## Characterization of neural activity

RFs were characterized during performance of the visually guided saccade task. An initial exploration covered 12–18 evenly spaced target locations at eccentricities of 4–15 degrees. After identifying the locations that elicited the strongest and weakest task-related responses, a new set of locations were selected around these initial two using integer degree values. The preferred location (i.e., the RF) and diametrically opposite site were selected from this refined grid.

All neurons included in the current study ($n = 51$ for CRDM task, $n = 56$ for urgent color-discrimination task) were significantly activated during performance of the urgent tasks, both in response to visual stimuli presented in their RF (window: 20:150 ms, aligned on targets on) as well as prior to saccades executed into their RF (window: −100:0 ms, aligned on saccade) relative to respective baseline measures. The visual and motor RFs of these neurons were highly consistent (Supplementary Fig. 13). In addition, all neurons included exhibited significant delay period activity in the visually- and/or memory-guided saccade tasks. For all such determinations, significance ($p < 0.01$) was calculated numerically via permutation tests[56] in which the two group labels (e.g., 'baseline' and 'response period') were randomly permuted. These physiological response properties (i.e., visual, delay period, and presaccadic activation) are characteristic of LIP neurons that project directly to saccade production centers[57], i.e., the superior colliculus. For the sampled populations, the median firing rate in response to a target appearing in the RF was 64 spikes/s (range was 5–173 spikes/s; rate computed in the time window 50–120 ms after target onset during the delayed saccade task).

Some additional neurons that were recorded and fully characterized (15 in the CRDM experiment, 26 in the color-based) were excluded from the studied samples for any of the following reasons: they had no significant visual or memory activity in the single-target tasks; they were not significantly activated presaccadically; or their spatial preference for contralateral/ipsilateral stimuli either was ambiguous or clearly flipped between different tasks. Importantly, though, except for small quantitative variations, all results were essentially identical with inclusion of all such neurons (Supplementary Fig. 8).

For each neuron, continuous firing rate traces, or spike density functions, were generated by aligning the recorded spike trains to relevant task events (e.g., cue onset, saccade onset), convolving them with a gaussian kernel ($\sigma = 15$ ms), and averaging across trials. Normalized population traces (as in panels a–c, e, f of Figs. 2 and 6) were generated by dividing each cell's response curve by its maximum firing rate value and then averaging across cells. For each cell, this maximum rate was calculated from the recorded urgent trials (motion- or color-based) and was used to normalize the population traces for all other tasks.

## ROC analyses and neurometric curves

The magnitude of spatial differentiation, or $S_{ROC}$, was used to quantify the degree to which LIP neurons were differentially activated in $S_{in}$ versus $S_{out}$ choices. This measure corresponds to the accuracy with which an ideal observer can classify data samples from two distributions (of responses in $S_{in}$ and $S_{out}$ trials, in this case), and is equivalent to the area under the receiver operating characteristic, or ROC, curve[58,59]. Values of 0.5 correspond to distributions that are indistinguishable (chance performance, full overlap), whereas values of 0 or 1 correspond to fully distinguishable distributions (perfect performance, no overlap). Here, $S_{ROC} > 0.5$ always indicates higher activity for saccades into the RF than away from the RF. Presaccadic $S_{ROC}$ values (Figs. 2g, h, 3b, e and 6g, h) were computed using spike counts measured prior to choice onset (window: −50:0 ms, aligned on saccade) and sorted according to trial outcome.

For the urgent tasks, continuous neurometric functions comparable to the behavioral tachometric curves (Figs. 2h and 6h) were generated by first pooling the data across neurons and then calculating $S_{ROC}$ as a function of rPT (bin width = 51 ms, shifted every 1 ms). The pooling involved two steps. First, the presaccadic spike counts of each neuron were centered by subtracting a constant, $\theta$, that was cell-specific, and then the centered spike counts from all the neurons were sorted into two groups, for $S_{in}$ and $S_{out}$ trials. The pooled $S_{ROC}$ compared responses from these two pooled distributions within each rPT bin (see Supplementary Fig. 14 for an example). For each neuron, the constant $\theta$ was equal to $(m_{in} + m_{out})/2$, where $m_{in}$ and $m_{out}$ are the mean spike counts for $S_{in}$ and $S_{out}$ trials. Other normalization schemes produced qualitatively similar trends. This procedure, pooling the data first and then computing $S_{ROC}$, generated more precise results than the reverse, i.e., first computing $S_{ROC}$ for each cell and then averaging across cells. However, the latter alternative produced qualitatively consistent results (Supplementary Fig. 11). We stress that, although the $S_{ROC}$ values that make up the neurometric curve vary with rPT, they were always based on spike counts measured just prior to the saccade.

For the non-urgent tasks (Figs. 2d and 6d), continuous $S_{ROC}$ values were again computed by dividing time into sliding bins (bin width = 50 ms, shifted every 1 ms). For each bin, the spikes counted for each neuron in each condition ($S_{in}$ and $S_{out}$ trials) were used to calculate that cell's $S_{ROC}$, and then values were averaged across cells. Pooling was unnecessary in this case because more trials were available per time bin, but the results with pooling were very similar. The onset of differentiation in the non-urgent tasks (Figs. 2d and 6d, arrows) was calculated as the earliest time point at which the mean $S_{ROC}$ was 2 SEs above chance level (0.5) and remained above thereafter.

## Statistical tests

Effect sizes for mean $S_{ROC}$ values were computed by bootstrapping[60,61]; that is, by repeatedly resampling the underlying data with replacement ($10^4$–$10^5$ iterations) and recomputing the mean $S_{ROC}$ each time. In Figs. 2g and 6g (insets), the resampling was over neurons; in Fig. 3b, e, it was over trials in the two pooled distributions (for $S_{in}$ and $S_{out}$ conditions). Effect sizes for other quantities (e.g., $\Delta c$ in Fig. 3c, f) were also calculated through bootstrapping. Having generated these effect-size distributions for any two conditions (e.g., correct vs. incorrect choices, or long vs. short rPTs), we could calculate from them a significance value for the mean difference. Instead, however, for any relevant comparison between two conditions, the $p$ value of the difference was calculated separately using a permutation test[56] for paired data or an equivalent resampling test for non-paired data, as these tests provide slightly more accurate and specific comparisons against the null hypothesis (of no difference between the distributions from which the two data sets originated). For example, to compare the mean $S_{ROC}$ for short- versus long-rPT trials (Figs. 2g and 6g, insets), we randomly permuted the 'short' and 'long' labels for each neuron and recomputed the difference between $S_{ROC}$ means $10^5$ times. Similarly, to compare the

**Article** https://doi.org/10.1038/s41467-022-32209-z

mean accuracy between two tachometric curves conditioned on neural activity (Fig. 3c, f, insets), we randomly reassigned the 'congruent' and 'incongruent' labels of the trials $10^5$ times, and each time, we recomputed the two tachometric curves and, from them, the difference in accuracy. The $p$ value was the fraction of iterations for which the difference was equal to or more extreme than that obtained from the original, non-permuted data. All reported significance values were calculated this way, via permutation or resampling tests (one-sided).

### Reporting summary

Further information on research design is available in the Nature Research Reporting Summary linked to this article.

## Data availability

The behavioral and presaccadic spike-count data that support the findings of this study are publicly available from the Zenodo repository, https://doi.org/10.5281/zenodo.6604002. Source data are provided with this paper.

## Code availability

Matlab scripts for reproducing analysis results are included as part of the shared data package at Zenodo, https://doi.org/10.5281/zenodo.6604002.

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

## Acknowledgements
We thank Alex Huk for constructive comments and Denise Anderson for expert laboratory assistance. The research was supported by the National Institutes of Health through grants R01EY025172 (to E.S. and T.R.S) and R01EY021228 (to E.S. and T.R.S) from the National Eye Institute. J.A.S. was supported by grant F31EY029154 (to J.A.S.) from the National Eye Institute and by Training Grant T32NS073553 from the National Institute for Neurological Disorders and Stroke.

## Author contributions
J.A.S., E.S. and T.R.S. designed the research, analyzed data, and wrote the manuscript; J.A.S. collected data.

## Competing interests
The authors declare no competing interests.
