## [Peer Review File · Nature Communications]

A conflict between spatial selection and evidence accumulation in area LIPREVIEWERS' COMMENTS

Reviewer #1 (Remarks to the Author):

The new edits have substantially improved the manuscript. The authors have thoroughly addressed my concerns and I have no further comments or questions. Fantastic work by the authors!

Reviewer #2 (Remarks to the Author):

I am satisfied with the authors' responses to my comments. I like their proposed explanation of the diffusion-like responses in LIP and I believe this paper will have a big impact. The new supplementary note is very helpful, and I really like the human data (Fig. R1). Surely this new data should be included as a supplementary figure?

Two final comments;

1. My original comment was, "In Figure 1, the label "Gap" spans the same space as "cue" leading the reader to believe that the Gap and cue times are the same." To which the authors responded,

"We are not quite sure about the source of confusion here, because the schematic shows the gap interval preceding the cue."

I am perplexed. In Figure 1c, there is a black square with dots on it, surely indicating the presence of dot motion. The text above the square says, "Gap". I am pretty sure this would convey to most readers that the epoch with dot motion is called the "Gap".

2. I still fear that the connection of these results to LIP inactivation data will be tenuous to readers. It is important to make this connection because many in the community were utterly baffled by the inactivation effects. I think that authors' explanation for the lack of inactivation effects is that LIP is not actually accumulating evidence, it just gives the false appearance of accumulating evidence because of a shift in attention from the dots to the choice targets (as explained in the revised text starting at line 247). Thus, inactivating LIP won't disrupt evidence accumulation because LIP is not doing that. The

authors state earlier in the discussion that their data accounts for inactivation data, but it may not be obvious that this is the explanation for why. I would make that connection clearer.

Reviewer #3 (Remarks to the Author):

The manuscript "A conflict between spatial selection and evidence accumulation in area LIP" by Seideman, Stanford and Salinas investigates the specific contribution neurons in area LIP make to decisions about motion direction in an urgent, curtailed motion discrimination task.

This is a revised submission of a manuscript I reviewed previously. The authors responded to the previous concerns and provided the requested additional information. I have no further questions.

Using an elegant task design, the authors provide new insights into the long-standing, pertinent issue of the key role LIP neurons might play in decision-making. The data are well presented, analysed and appear robust. The idea that conflicting results about LIP's role could be explained from the perspective of attentional allocation is not entirely new, but the evidence underpinning the conclusions is striking and of key significance to the field.

Response to reviewers

We are very grateful to the three reviewers for taking the time to again go over our manuscript. We sincerely appreciate their input.

Reviewer #1

The new edits have substantially improved the manuscript. The authors have thoroughly addressed my concerns and I have no further comments or questions. Fantastic work by the authors!

Thanks for the encouraging words!

Reviewer #2

I am satisfied with the authors' responses to my comments. I like their proposed explanation of the diffusion-like responses in LIP and I believe this paper will have a big impact. The new supplementary note is very helpful, and I really like the human data (Fig. R1). Surely this new data should be included as a supplementary figure?

We are very glad to hear this. Indeed, the Supplementary Note makes an important clarification that we had not articulated before; thanks for pointing us in that direction.

We agree that the human data are an excellent complement to the article, but because they are preliminary and part of an ongoing experiment that we plan to publish separately, we would rather not include them as an official part of this publication.

Two final comments;

1. My original comment was, "In Figure 1, the label "Gap" spans the same space as "cue" leading the reader to believe that the Gap and cue times are the same." To which the authors responded,

"We are not quite sure about the source of confusion here, because the schematic shows the gap interval preceding the cue."

I am perplexed. In Figure 1c, there is a black square with dots on it, surely indicating the presence of dot motion. The text above the square says, "Gap". I am pretty sure this would convey to most readers that the epoch with dot motion is called the "Gap".

Sorry about the confusion; we understand now. . . . In the schematic, the black squares correspond to events and the gray lines to intervals between events (RT, Gap, rPT). This is now indicated in the caption. We also changed the layout of the interval lines to make the distinction more obvious.

2. I still fear that the connection of these results to LIP inactivation data will be tenuous to readers. It is important to make this connection because many in the community were utterly baffled by the inactivation effects. I think that authors' explanation for the lack of inactivation effects is that LIP is not actually accumulating evidence, it just gives the false appearance of accumulating evidence because of a shift in attention from the dots to the choice targets (as explained in the revised text starting at line 247). Thus, inactivating LIP won't disrupt evidence accumulation because LIP is

not doing that. The authors state earlier in the discussion that their data accounts for inactivation data, but it may not be obvious that this is the explanation for why. I would make that connection clearer.

Thanks for pointing this out. Indeed, our explanation for the inactivation results is very much as the reviewer indicated. Now, in the Discussion, the inactivation experiments are mentioned after having unpacked our interpretation of the CRDM results based on spatial attention, at which point we explain in more detail the connection between the two.

Reviewer #3

The manuscript “A conflict between spatial selection and evidence accumulation in area LIP” by Seideman, Stanford and Salinas investigates the specific contribution neurons in area LIP make to decisions about motion direction in an urgent, curtailed motion discrimination task.

This is a revised submission of a manuscript I reviewed previously. The authors responded to the previous concerns and provided the requested additional information. I have no further questions.

Using an elegant task design, the authors provide new insights into the long-standing, pertinent issue of the key role LIP neurons might play in decision-making. The data are well presented, analysed and appear robust. The idea that conflicting results about LIP’s role could be explained from the perspective of attentional allocation is not entirely new, but the evidence underpinning the conclusions is striking and of key significance to the field.

Thanks so much!